# Vulnerabilities in Video Quality Assessment Models: The Challenge of Adversarial Attacks

**Ao-Xiang Zhang**      **Yu Ran**      **Weixuan Tang**[*]      **Yuan-Gen Wang**[*]

Guangzhou University, China
{zax, ranyu}@e.gzhu.edu.cn, {tweix, wangyg}@gzhu.edu.cn

## Abstract

No-Reference Video Quality Assessment (NR-VQA) plays an essential role in improving the viewing experience of end-users. Driven by deep learning, recent NR-VQA models based on Convolutional Neural Networks (CNNs) and Transformers have achieved outstanding performance. To build a reliable and practical assessment system, it is of great necessity to evaluate their robustness. However, such issue has received little attention in the academic community. In this paper, we make the first attempt to evaluate the robustness of NR-VQA models against adversarial attacks, and propose a patch-based random search method for black-box attack. Specifically, considering both the attack effect on quality score and the visual quality of adversarial video, the attack problem is formulated as misleading the estimated quality score under the constraint of just-noticeable difference (JND). Built upon such formulation, a novel loss function called Score-Reversed Boundary Loss is designed to push the adversarial video's estimated quality score far away from its ground-truth score towards a specific boundary, and the JND constraint is modeled as a strict $L_2$ and $L_\infty$ norm restriction. By this means, both white-box and black-box attacks can be launched in an effective and imperceptible manner. The source code is available at https://github.com/GZHU-DVL/AttackVQA.

## 1 Introduction

In recent years, the service of "we-media" has shown an explosive growth. It is reported that Facebook can produce approximately 4 billion video views per day [1]. Nevertheless, the storage and transmission of such volumes of video data pose a significant challenge to video service providers [2]. To address this challenge, video coding is employed to balance the tradeoff between coding efficiency and video quality. Therefore, Video Quality Assessment (VQA) has become a prominent research topic. In general, according to whether pristine video is applied as a reference, VQA can be divided into Full-Reference VQA (FR-VQA), Reduced-Reference VQA (RR-VQA), and No-Reference VQA (NR-VQA). Recently, Convolutional Neural Networks (CNNs) and Transformers have made great progress in NR-VQA. However, according to recent studies, Deep Neural Networks (DNNs) show vulnerability against adversarial examples [19]. To build a reliable and practical assessment system, it is of great necessity to evaluate the robustness of the NR-VQA models against adversarial attacks.

The threat of adversarial attacks has been studied in the task of image classification. Szegedy *et al.* [4] first found that an image injected with a small amount of imperceptible adversarial perturbations can mislead DNNs with high confidence. Moosav-DezFool *et al.* [5] suggested that universal perturbations can achieve attack effect on different samples. Athalye *et al.* [6] showed that adversarial examples can be applied in physical world. Since then, numerous adversarial attacks on CNNs and Transformers have been proposed [7; 8; 9; 24; 32].

---

[*]denotes corresponding author.

37th Conference on Neural Information Processing Systems (NeurIPS 2023).

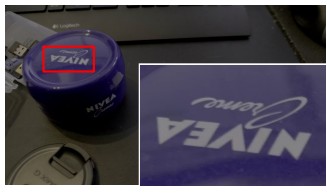 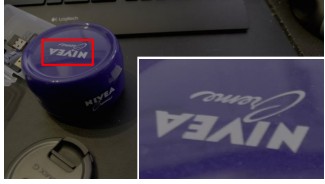 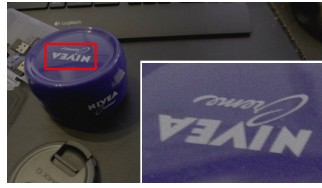

| Original | $L_2 = 3 / 255,\ L_\infty = 5 / 255$ | $L_2 = 8 / 255,\ L_\infty = 10 / 255$ |
|---|---|---|

Figure 1: The original video frame and the perturbed ones constrained by the $L_2$ and $L_\infty$ norm.

Compared with its rapid development in the classification task, there have been few investigations of adversarial attacks in Quality Assessment (QA), including Image Quality Assessment (IQA) and Video Quality Assessment (VQA). Two pioneering works [44; 50] were the closest to this topic in scope. [44] first investigated the adversarial attack on IQA models. It optimized the perturbations with propagated gradients by NR-IQA models. Such perturbations were added to the original image to yield adversarial images. Multiple adversarial images could be generated under the constraint of different Lagrange multipliers, and those adversarial images added with the most perceptually invisible perturbation were selected according to human subjective experiments. This can be understood as a constraint limiting the perturbations below the just-noticeable difference (JND) threshold. [50] designed a universal perturbation as the fixed perturbation trained by a given IQA/VQA model to increase its output scores. Despite [44] and [50] have made early attempt of adversarial attack on IQA/VQA model, there still remain some important issues to be solved. Firstly, since human subjective experiments are both time-consuming and labor-intensive, the test set of [44] includes only 12 images, which is too small to be practical. Secondly, [50] aims to merely increase the estimated quality score outputted by the target model, which is not in line with the definition of adversarial attack. Thirdly, [50] cannot control the visual quality during optimization, and thus the adversarial videos suffer from obvious artifacts. Fourthly, the adversarial attacks in both [44] and [50] are ineffective in more practical black-box scenarios.

To address the above problems, this paper comprehensively investigates the robustness of NR-VQA models by adversarial attacks. Firstly, adversarial attack on NR-VQA model is mathematically modelled. The target NR-VQA model takes in the adversarial video and outputs the estimated quality score. Meanwhile, we define the anchor quality score as far away from its mean opinion score (MOS) in a specific direction. The optimization goal of such attack problem is to mislead the estimated quality score by means of pulling in the distance between the estimated and anchor quality scores under the constraint of restricting the distortion between the adversarial and original videos below the JND threshold. Specifically, the Score-Reversed Boundary Loss is designed to push the adversarial video's estimated quality score far away from its MOS towards a specific boundary. And limiting the $L_2$ and $L_\infty$ norm of the perturbations below a rather strict threshold can be interpreted as a kind of JND constraint. In this paper, unless otherwise specified, the $L_2$ norm indicates the pixel-level $L_2$ norm averaged on the whole video. An example of applying the constraint of $L_2$ and $L_\infty$ norm is given in Fig. 1. It can be observed that by means of limiting the $L_2$ and $L_\infty$ norm of perturbations within a rather strict threshold of 3/255 and 5/255, the perturbations are imperceptible to human eyes, which can be considered to be below the JND threshold. The contributions of this work are summarized as follows:

- Adversarial attacks on NR-VQA models are formulated as misleading the estimated quality score by means of pulling in the distance between the estimated and anchor quality scores under the JND constraint. To the best of our knowledge, this is the first work to apply adversarial attacks to investigate the robustness of NR-VQA models under black-box setting.

- A novel loss function called Score-Reversed Boundary Loss is proposed to push the adversarial video's estimated quality score far away from its MOS towards a specific boundary. By means of minimizing such loss function, the generated adversarial video can effectively disable the NR-VQA models.

- Adversarial attacks on NR-VQA models are implemented under white-box and black-box settings. Furthermore, a patch-based random search method is designed for black-box attack, which can significantly improve the query efficiency in terms of both the spatial and temporal domains.

## 2 Related work

### 2.1 NR-VQA models

With the rapid development of deep learning techniques, NR-VQA models based on CNNs and Transformers have received tremendous attention. Li *et al.* [25] extracted the 3D shearlet transform features from video blocks. Afterwards, logistic regression and CNN were simultaneously applied to exaggerate the discriminative parts of the features and predict the quality score. Li *et al.* [26] utilized ResNet [27] to extract features for each frame within video, which were fed into GRU [28] to construct the long-term dependencies in temporal domain. You and Korhonen [29] employed LSTM [30] to predict quality score based on the features extracted from small video cubic clips. Zhang *et al.* [40] reorganized the connections of human visual system, and comprehensively evaluated the quality score from spatial and temporal domains. Li *et al.* [31] transferred the knowledge from IQA to VQA, and applied SlowFast [34] to obtain the motion patterns of videos.

Xing *et al.* [35] were the first to introduce Transformer [38] into NR-VQA to implement space-time attention. Wu *et al.* [36] designed a spatial-temporal distortion extraction module, wherein the video Swin Transformer [42] was utilized to extract multi-level spatial-temporal features. Li and Yang [37] proposed a hierarchical Transformer strategy, wherein one Transformer was utilized to update the frame-level quality embeddings, and another Transformer was utilized to generate the video-level quality embeddings. To reduce the computational complexity of Transformers, Wu *et al.* [41] segmented the video frames into smaller patches, which were partially concatenated into fragments and fed into video Swin Transformer [42].

### 2.2 Adversarial attacks in classification

According to the knowledge of the attacker, adversarial attacks can be roughly divided into white-box and black-box attacks. In general, most white-box attacks rely on the propagated gradients of the target model. Goodfellow *et al.* [16] suggested that DNNs were susceptible to analytical perturbations due to their linear behaviors, and proposed the Fast Gradient Sign Method (FGSM). On this basis, Kurakin *et al.* [17] launched the attack via several steps, and clipped the pixel values within the $\epsilon$-neighbourhood after each step. Similarly, Madry *et al.* [20] proposed a multi-step variation of FGSM called Projected Gradient Descent (PGD), wherein the updated adversarial example was projected on the $L_{\infty}$ norm in each PGD iteration. Carlini and Wagner [3] formulated the adversarial attack as an optimization problem with different objective functions, which aimed to find adversarial examples with minimum perturbations. Papernot *et al.* [18] created adversarial examples by iteratively modifying the most significant pixel based on the adversarial saliency map.

However, in real-world scenarios, the gradients of the model may not be accessible to the attacker. Therefore, the black-box setting is a more practical attack scenario. Papernot *et al.* [21] performed a black-box attack by training a local surrogate model, and then used this surrogate model to generate adversarial examples that could mislead the target model. Chen *et al.* [22] proposed the Zeroth Order Optimization (ZOO) based attack to estimate the gradients of the target model. To improve the query efficiency of ZOO, Ilyas *et al.* [23] replaced the conventional finite differences method with natural evolution strategy in gradient estimation. Guo *et al.* [24] proposed a simple black-box attack, which iteratively injected perturbations that could lead to performance degradation of target model in a greedy way. Xiao *et al.* [43] proposed AdvGAN to generate perturbations via feed-forward process, which could accelerate the generation of adversarial examples in the inference stage.

## 3 Proposed method

### 3.1 Problem formulation

In this part, the adversarial attack on NR-VQA model is mathematically modelled as

$$\underset{\mathbf{I}_x^{\mathrm{adv}}}{\mathrm{argmin}} \, \mathcal{L}\left(f\left(\left\{\mathbf{I}_x^{\mathrm{adv}}\right\}_{x=1}^{X}\right), b\left(\left\{\mathbf{I}_x\right\}_{x=1}^{X}\right)\right), \, s.t. \, \mathcal{D}\left(\left\{\mathbf{I}_x\right\}_{x=1}^{X}, \left\{\mathbf{I}_x^{\mathrm{adv}}\right\}_{x=1}^{X}\right) \leq \mathrm{JND}, \quad (1)$$

where $X$ denotes the number of frames within a video, and $\left\{\mathbf{I}_x^{\mathrm{adv}}\right\}_{x=1}^{X}$ and $\left\{\mathbf{I}_x\right\}_{x=1}^{X}$ denote the adversarial video and original video, respectively. $f(\cdot)$ denotes the estimated quality score outputted

---

**Algorithm 1** White-box attack on NR-VQA model

---

1: **Require:** Estimated score by target NR-VQA model $f(\cdot)$, anchor score $b(\cdot)$, number of frames optimized in one round $T$, number of iterations in one round $K$, and step size $\beta$

2: **Input:** Original video $\{\mathbf{I}_x\}_{x=1}^{X}$

3: **Output:** Adversarial video $\{\mathbf{I}_x^{\text{adv}}\}_{x=1}^{X}$

4: **for** $x = 0$ to $X$ by $T$ **do**         *// The optimization process contains $\lfloor X/T \rfloor$ rounds.*

5:      $\{\mathbf{A}_x\}_x^{x+T} \leftarrow$ Initialize the perturbations from a discrete set $\{-1/255, 0, 1/255\}$

6:      $\{\mathbf{I}_x^{\text{adv}}\}_x^{x+T} \leftarrow \{\mathbf{I}_x\}_x^{x+T} + \{\mathbf{A}_x\}_x^{x+T}$         *// Initialize the adversarial video.*

7:      **for** $k = 0$ to $K$ **do**         *// One round contains $K$ iterations.*

8:          $g \leftarrow \nabla_{\{\mathbf{I}_x^{\text{adv}}\}_x^{x+T}} \mathcal{L}_{srb}\left(f\left(\{\mathbf{I}_x^{\text{adv}}\}_x^{x+T}\right), b\left(\{\mathbf{I}_x\}_{x=1}^{X}\right)\right)$      *//Compute the direction.*

9:          $\{\mathbf{I}_x^{\text{adv}}\}_x^{x+T} \leftarrow \{\mathbf{I}_x^{\text{adv}}\}_x^{x+T} + \beta \cdot g$         *// Update the adversarial video.*

10:          $\{\mathbf{I}_x^{\text{adv}}\}_x^{x+T} \leftarrow$ Limit the pixel-level $L_2$ and $L_\infty$ norm of perturbations

---

by the target NR-VQA model, and $b(\cdot)$ denotes the anchor quality score which is far away from its MOS in a specific direction. $\mathcal{L}(\cdot)$ is the loss function that measures the distance between the estimated and anchor quality scores. The closer the distance between the estimated and anchor quality scores, the further the distance between the estimated quality score and its MOS, and the stronger the effect of adversarial attack on misleading the estimated quality score. $\mathcal{D}(\cdot)$ is the distortion between the original and adversarial videos. Restricting the distortion below the JND threshold indicates that the perturbations are imperceptible to human eyes. By means of minimizing the loss function under the JND constraint, the adversarial attack on NR-VQA model can be launched in an effective and undetectable manner.

To implement the adversarial attack according to Eq. (1), it is essential to design the loss function $\mathcal{L}$. Although Mean Squared Error (MSE) Loss has been widely applied in regression problem, it is not suitable for adversarial attack on NR-VQA model. The reason is that a well-performed adversarial attack on NR-VQA model should possess the capability of leading the estimated quality score far away from its MOS in a specific direction, *i.e.,* misleading the NR-VQA model to assign high-quality score to low-quality video, and vice versa. Although applying MSE Loss may push the estimated quality score away from the MOS, it cannot control the leading direction. In this paper, a loss function called Score-Reversed Boundary Loss ($\mathcal{L}_{srb}$) is proposed, which is formulated as

$$\mathcal{L}_{srb} = \left| f\left(\{\mathbf{I}_x^{\text{adv}}\}_{x=1}^{X}\right) - b\left(\{\mathbf{I}_x\}_{x=1}^{X}\right) \right|, \tag{2}$$

where

$$b\left(\{\mathbf{I}_x\}_{x=1}^{X}\right) = \text{boundary} = \begin{cases} 0, & \text{if } \{\mathbf{I}_x\}_{x=1}^{X} \text{ is high quality,} \\ 1, & \text{if } \{\mathbf{I}_x\}_{x=1}^{X} \text{ is low quality,} \end{cases} \tag{3}$$

where the high or low quality of a video in Eq. (3) can be determined by its MOS in practice, and the boundaries of 0 and 1 are regarded as the anchor quality scores.

Besides, the perturbation distortion $\mathcal{D}$ between the original and adversarial videos is another issue to be addressed. In general, limiting $\mathcal{D}$ below the JND threshold indicates that such perturbations are invisible to human eyes. However, JND is empirically determined by many factors, including brightness, contrast between the background, and individual visual sensitivity. Therefore, some fields convert it into certain objective metrics for simple modeling, such as PSNR in video coding [10], SSIM in image/video enhancement [11], and $L_p$ norm in adversarial attacks [13]. Zhang *et al.* [44] introduced human subjective experiments to judge whether the perturbation distortion is below the JND threshold. However, such human subjective experiments are both time-consuming and labor-intensive. In this paper, we refer to the works in adversarial examples to construct the JND constraint [12; 13; 14]. Considering a pair of original and adversarial videos, the problem of constraining the distortion below the JND threshold can be converted into the problem of restricting their pixel-level $L_2$ and $L_\infty$ norm to be lower than a rather strict threshold. Consequently, the optimization based on subjective JND can be automatically performed by the objective metrics.

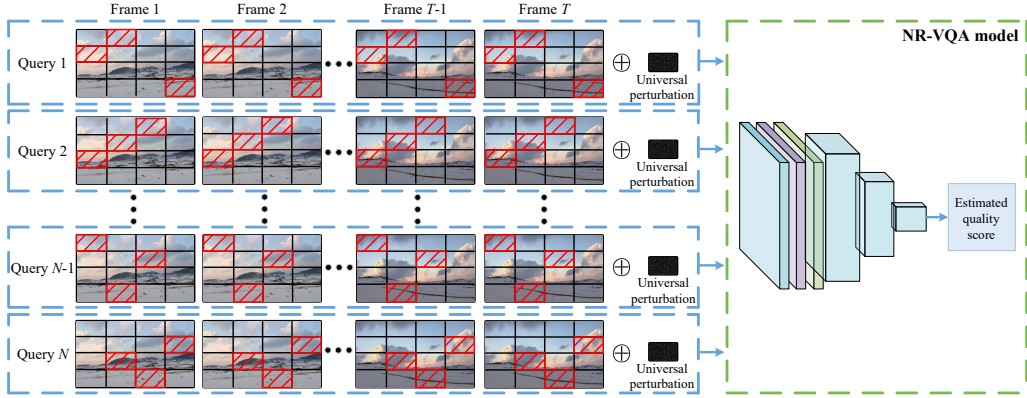

Figure 2: Illustration of the proposed patch-based random search method for black-box attack.

## 3.2 White-box adversarial attack

In the case of white-box attack, the parameters of the target model are accessible to the attacker. Therefore, the steepest descent method can be applied to minimize the Score-Reversed Boundary Loss in Eq. (2). The process of generating an adversarial video has $\lfloor X/T \rfloor$ rounds, wherein $T$ denotes the number of frames optimized in one round together. At the beginning of each round, the adversarial videos are generated by means of adding the original videos with initialized perturbations denoted as $\{\mathbf{A}_x\}_x^{x+T}$, whose element is independently sampled from a discrete set $\{-1/255, 0, 1/255\}$. Afterwards, $K$ iterations are applied in this round to optimize the perturbations, and the detailed steps of one iteration are as follows. Firstly, the obtained adversarial videos are fed into the target NR-VQA model, and the steepest descent direction of $\mathcal{L}_{srb}$ can be calculated to optimize the perturbations. Secondly, the projected gradient descent method is utilized to limit the pixel-level $L_2$ and $L_\infty$ norm of the perturbations within 1/255, which can be regarded as a kind of strict JND constraint. The pseudo-code of generating adversarial example for NR-VQA model under white-box setting is given in Algorithm 1.

## 3.3 Black-box adversarial attack

Compared with white-box attack, black-box attack is a more practical attack scenario, wherein only the final output of the model can be queried to optimize the adversarial examples. Note that the number of queries is an important evaluation metric in black-box attack. However, black-box attack on NR-VQA model pose great challenges from the following two aspects. Firstly, in most existing outstanding black-box attacks, the number of queries is positively correlated with the resolution of the video frame. As NR-VQA models are widely applied to 1080p and 2160p high-resolution datasets such as LIVE-VQC [15], YouTube-UGC [45], and LSVQ [46], its query resources may be quite expensive. Secondly, different frames within one video have different content characteristics. Therefore, to obtain better attack effect, each frame should be generated with a specific perturbation. However, one video may consist of hundreds of frames, which can lead to larger number of queries.

To reduce the query resources from the spatial and temporal perspective under black-box attack, we design a patch-based random search method, as illustrated in Fig. 2. In this part, the height and width of a video frame are denoted as $H$ and $W$, respectively. The height and width of a patch are denoted as $h$ and $w$, respectively. Each frame is divided into red, green, and blue channels, and each channel is further split into non-overlapping $\lfloor \frac{H}{h} \rfloor \times \lfloor \frac{W}{w} \rfloor$ patches. In this way, each frame contains $\lfloor \frac{H}{h} \rfloor \times \lfloor \frac{W}{w} \rfloor \times 3$ patches.

Like the white-box attack, we suppose that the process of generating an adversarial video has $\lfloor X/T \rfloor$ rounds. In one round, $N$ queries are performed. The detailed steps of the optimization process in the $n$-th query are as follows. Firstly, for each of the $T$ frames, $Z$ patches, which are located in the same positions within these $T$ frames, are randomly selected to be perturbed. Note that in each round of the attack, each of the $Z$ patches would be selected once and only once, and all patches would be selected in $N$ queries. Therefore, we can obtain $N \times Z = \lfloor \frac{H}{h} \rfloor \times \lfloor \frac{W}{w} \rfloor \times 3$. For the convenience of

---

**Algorithm 2** Black-box attack on NR-VQA model

---

1: **Require:** Estimated score by target NR-VQA model $f(\cdot)$, anchor score $b(\cdot)$, number of frames optimized in one round $T$, number of queries in one round $N$, perturbation magnitude $\gamma$, size of video frame $H \times W$, and size of patch $h \times w$

2: **Input:** Original video $\{\mathbf{I}_x\}_{x=1}^{X}$

3: **Output:** Adversarial video $\{\mathbf{I}_x^{\mathrm{adv}}\}_{x=1}^{X}$

4: **for** $x = 0$ to $X$ by $T$ **do**                    *// The optimization process contains $\lfloor X/T \rfloor$ rounds.*

5:     $\{\mathbf{I}_x^{\mathrm{adv}}\}_x^{x+T} \leftarrow \{\mathbf{I}_x\}_x^{x+T}$                    *// Initialize the adversarial video.*

6:     $R \leftarrow \mathcal{L}_{srb}\left(f\left(\{\mathbf{I}_x^{\mathrm{adv}}\}_x^{x+T}\right), b\left(\{\mathbf{I}_x\}_{x=1}^{X}\right)\right)$

7:     **for** $n = 0$ to $N$ **do**                    *// One round contains $N$ queries.*

8:         $Z \leftarrow \max\left(\lfloor \lfloor \frac{H}{h} \rfloor \times \lfloor \frac{W}{w} \rfloor \times 3/N \rfloor, 1\right)$         *// Select $Z$ patches to perturb in one frame.*

9:         $\mathbf{p}_n \leftarrow$ Encode information of the positions of the selected patches

10:         $\mathbf{m}_n \leftarrow$ Generate universal perturbation map from a discrete set $\{-\gamma, +\gamma\}$

11:         **for** operation $\in \{-, +\}$ **do**

12:             $\{\tilde{\mathbf{I}}_x^{\mathrm{adv}}\}_x^{x+T} \leftarrow$ Selected patches are injected with $\mathbf{m}_n$ according to given operation

13:             $R' \leftarrow \mathcal{L}_{srb}\left(f\left(\{\tilde{\mathbf{I}}_x^{\mathrm{adv}}\}_x^{x+T}\right), b\left(\{\mathbf{I}_x\}_{x=1}^{X}\right)\right)$         *// Evaluate the attack effect.*

14:             **if** $R' < R$ **then**

15:                 $\{\mathbf{I}_x^{\mathrm{adv}}\}_x^{x+T} \leftarrow \{\tilde{\mathbf{I}}_x^{\mathrm{adv}}\}_x^{x+T}$         *// Update the adversarial video.*

16:                 $R \leftarrow R'$

17:                 **break**

---

illustration, the information of the positions of the selected patches is encoded into a vector denoted as $\mathbf{p}_n^{K \times 1}$, where $K = \lfloor \frac{H}{h} \rfloor \times \lfloor \frac{W}{w} \rfloor \times 3$. Each element in $\mathbf{p}_n$ corresponds to a specific patch within the frame, and the element equals to 1 if such patch is perturbed, and 0 otherwise. Note that each of the $T$ frames optimized in one round shares the same $\mathbf{p}_n$. Secondly, for these $T \times Z$ patches to be perturbed, a universal perturbation map $\mathbf{m}_n$ is generated, wherein its dimension is the same as the dimension of the patch, and its element is independently sampled from a discrete set $\{-\gamma, +\gamma\}$. Thirdly, the selected $T \times Z$ patches are subtracted or added with the universal perturbation map, and the generated adversarial video is fed into the target NR-VQA model. Such perturbations would be kept if the Score-Reversed Boundary Loss in Eq. (2) decreases, and abandoned otherwise. Following the above three steps, an adversarial video with $X$ frames can be generated. The pseudo-code of generating adversarial example for NR-VQA model under black-box setting is given in Algorithm 2.

By means of applying the patch-based random search method, the constraint of JND can be formulated under $L_\infty$ or pixel-level $L_2$ norm. Firstly, the $L_\infty$ norm between the original and adversarial videos is $\gamma$, which is the maximum modification range of each element in the adversarial video. Secondly, the frame-level $L_2$ norm between the original and adversarial videos can be formulated as

$$\sqrt{H \times W \times 3 \times L_2^2} = \|\sum_{n=1}^{N} \gamma \mathbf{p}_n\|_2 = \gamma \|\sum_{n=1}^{N} \mathbf{p}_n\|_2 \leq \gamma \sqrt{N \times Z \times h \times w}, \tag{4}$$

where the last inequality holds as different patches are selected once and only once in different rounds, and the last equality holds when all patches are injected with perturbations. And $\gamma$ is set as 5/255 in black-box attack. Therefore, the pixel-level $L_2$ norm can be deduced that

$$L_2 \leq \sqrt{\frac{\gamma^2 \times N \times Z \times h \times w}{H \times W \times 3}} = \sqrt{\frac{\gamma^2 \times \frac{H}{h} \times \frac{W}{w} \times 3 \times h \times w}{H \times W \times 3}} = \gamma. \tag{5}$$

## 4 Experiments

### 4.1 Experimental setup

**NR-VQA models and datasets:** Four representative NR-VQA models are tested in the experiments, including VSFA [26], MDTVSFA [39], TiVQA [49], and BVQA-2022 [31]. The experiments are

Table 1: Performance evaluations under white-box setting with $L_2$ norm constraint. Here, the performance before attack is marked in gray values within parentheses.

(a) Performance evaluations on VSFA [26] and MDTVSFA [39].

| Metric | SRCC | | PLCC | | $R$ defined in Eq. (6) | |
|---|---|---|---|---|---|---|
| Model | VSFA | MDTVSFA | VSFA | MDTVSFA | VSFA | MDTVSFA |
| KoNViD-1k | -0.7198 (0.7882) | -0.7380 (0.8003) | -0.8135 (0.8106) | -0.8212 (0.8074) | -0.2507 | -0.0368 |
| LIVE-VQC | -0.7518 (0.7665) | -0.7825 (0.7908) | -0.7322 (0.7506) | -0.7729 (0.8091) | -0.4963 | -0.3121 |
| YouTube-UGC | -0.7071 (0.7492) | -0.7223 (0.7683) | -0.8018 (0.7721) | -0.7939 (0.7950) | -0.3585 | -0.2883 |
| LSVQ-Test | -0.6723 (0.7865) | -0.6623 (0.8136) | -0.6765 (0.7972) | -0.7906 (0.8125) | 0.2692 | 0.4252 |
| LSVQ-Test-1080p | -0.7847 (0.7040) | -0.7573 (0.7114) | -0.7830 (0.6868) | -0.7885 (0.7009) | 0.0527 | 0.2589 |

(b) Performance evaluations on TiVQA [49] and BVQA-2022 [31].

| Metric | SRCC | | PLCC | | $R$ defined in Eq. (6) | |
|---|---|---|---|---|---|---|
| Model | TiVQA | BVQA-2022 | TiVQA | BVQA-2022 | TiVQA | BVQA-2022 |
| KoNViD-1k | -0.7212 (0.8046) | -0.7371 (0.8427) | -0.8204 (0.8377) | -0.8233 (0.8469) | -0.2472 | -0.0579 |
| LIVE-VQC | -0.6643 (0.8179) | -0.6705 (0.8698) | -0.7352 (0.8067) | -0.6834 (0.8457) | -0.2144 | -0.0042 |
| YouTube-UGC | -0.6179 (0.7907) | -0.6696 (0.8145) | -0.7356 (0.8177) | -0.6177 (0.8500) | 0.0574 | 0.1118 |
| LSVQ-Test | -0.7457 (0.8331) | -0.7679 (0.8613) | -0.7898 (0.8343) | -0.8063 (0.8531) | 0.0248 | 0.0693 |
| LSVQ-Test-1080p | -0.7372 (0.7478) | -0.7088 (0.7875) | -0.7249 (0.7670) | -0.7569 (0.7799) | 0.1979 | 0.2037 |

Table 2: Performance evaluations under white-box setting with $L_\infty$ norm constraint. Here, the performance before attack is marked in gray values within parentheses.

(a) Performance evaluations on VSFA [26] and MDTVSFA [39].

| Metric | SRCC | | PLCC | | $R$ defined in Eq. (6) | |
|---|---|---|---|---|---|---|
| Model | VSFA | MDTVSFA | VSFA | MDTVSFA | VSFA | MDTVSFA |
| KoNViD-1k | -0.7155 (0.7882) | -0.7296 (0.8003) | -0.8139 (0.8106) | -0.8219 (0.8074) | -0.2521 | -0.0419 |
| LIVE-VQC | -0.7526 (0.7665) | -0.7766 (0.7908) | -0.7323 (0.7506) | -0.7720 (0.8091) | -0.4968 | -0.3129 |
| YouTube-UGC | -0.7013 (0.7492) | -0.7238 (0.7683) | -0.8025 (0.7721) | -0.7946 (0.7950) | -0.3669 | -0.2974 |
| LSVQ-Test | -0.6601 (0.7865) | -0.6569 (0.8136) | -0.6824 (0.7972) | -0.7949 (0.8125) | 0.2551 | 0.4086 |
| LSVQ-Test-1080p | -0.7864 (0.7040) | -0.7438 (0.7114) | -0.7833 (0.6868) | -0.7878 (0.7009) | 0.0528 | 0.2533 |

(b) Performance evaluations on TiVQA [49] and BVQA-2022 [31].

| Metric | SRCC | | PLCC | | $R$ defined in Eq. (6) | |
|---|---|---|---|---|---|---|
| Model | TiVQA | BVQA-2022 | TiVQA | BVQA-2022 | TiVQA | BVQA-2022 |
| KoNViD-1k | -0.7419 (0.8046) | -0.7265 (0.8427) | -0.8230 (0.8377) | -0.8224 (0.8469) | -0.2521 | -0.0708 |
| LIVE-VQC | -0.6833 (0.8179) | -0.6712 (0.8698) | -0.7425 (0.8067) | -0.6867 (0.8457) | -0.2214 | -0.0229 |
| YouTube-UGC | -0.6034 (0.7907) | -0.6920 (0.8145) | -0.7199 (0.8177) | -0.6889 (0.8500) | 0.0479 | 0.0817 |
| LSVQ-Test | -0.7591 (0.8331) | -0.7769 (0.8613) | -0.7935 (0.8343) | -0.8093 (0.8531) | 0.0178 | 0.0533 |
| LSVQ-Test-1080p | -0.7351 (0.7478) | -0.7128 (0.7875) | -0.7442 (0.7670) | -0.7613 (0.7799) | 0.1875 | 0.1876 |

conducted on mainstream video datasets including KoNViD-1k [48], LIVE-VQC [15], YouTube-UGC [45], and LSVQ [46]. Specifically, LSVQ is further divided into two datasets according to their resolutions. Therefore, five datasets are applied in the experiments. For each of these five datasets, 50 videos are randomly selected to evaluate the effectiveness of adversarial attacks.

**Evaluation metrics:** The performance of NR-VQA model is evaluated by Spearman Rank order Correlation Coefficient (SRCC) and Pearson's Linear Correlation Coefficient (PLCC). Specifically, SRCC is used to measure the monotonic relationship between the predicted quality scores and the ground-truth quality scores, while PLCC is used to measure the accuracy of the predictions. Note that both SRCC and PLCC are evaluated on a batch of videos. However, in real attack scenario, the attacker may merely launch the adversarial attack on a few samples. In this case, the change range of SRCC and PLCC could be rather small, and thus these two metrics may not accurately reflect the attack effect. Therefore, we follow [44] to define a metric on a single video, which is calculated as

Table 3: Performance evaluations under black-box setting. Here, the performance before attack is marked in gray values within parentheses.

(a) Performance evaluations on VSFA [26] and MDTVSFA [39].

| Metric | SRCC | | PLCC | | $R$ defined in Eq. (6) | |
|---|---|---|---|---|---|---|
| Model | VSFA | MDTVSFA | VSFA | MDTVSFA | VSFA | MDTVSFA |
| KoNViD-1k | -0.0305 (0.7882) | 0.0261 (0.8003) | 0.0586 (0.8106) | 0.1235 (0.8074) | 1.6573 | 1.7587 |
| LIVE-VQC | -0.1605 (0.7665) | -0.0074 (0.7908) | -0.0132 (0.7506) | 0.0807 (0.8091) | 1.3233 | 1.4391 |
| YouTube-UGC | -0.2231 (0.7492) | -0.0646 (0.7683) | -0.1017 (0.7721) | 0.0532 (0.7950) | 1.5499 | 1.5080 |
| LSVQ-Test | 0.2386 (0.7865) | -0.0706 (0.8136) | 0.2413 (0.7972) | -0.0384 (0.8125) | 2.3545 | 2.1352 |
| LSVQ-Test-1080p | 0.2351 (0.7040) | -0.1339 (0.7114) | 0.2840 (0.6868) | -0.0288 (0.7009) | 2.1839 | 1.8592 |

(b) Performance evaluations on TiVQA [49] and BVQA-2022 [31].

| Metric | SRCC | | PLCC | | $R$ defined in Eq. (6) | |
|---|---|---|---|---|---|---|
| Model | TiVQA | BVQA-2022 | TiVQA | BVQA-2022 | TiVQA | BVQA-2022 |
| KoNViD-1k | -0.4992 (0.8046) | 0.1978 (0.8427) | -0.4099 (0.8377) | 0.2215 (0.8469) | 1.2458 | 2.0480 |
| LIVE-VQC | -0.0762 (0.8179) | 0.4571 (0.8698) | 0.0452 (0.8067) | 0.4778 (0.8457) | 1.3234 | 2.2061 |
| YouTube-UGC | -0.1357 (0.7907) | 0.3951 (0.8145) | -0.0087 (0.8177) | 0.4306 (0.8500) | 1.3478 | 2.1072 |
| LSVQ-Test | -0.3343 (0.8331) | 0.3355 (0.8613) | -0.2161 (0.8343) | 0.4240 (0.8531) | 1.8086 | 2.2269 |
| LSVQ-Test-1080p | -0.1414 (0.7478) | 0.3917 (0.7875) | -0.0343 (0.7670) | 0.4678 (0.7799) | 1.3327 | 2.0796 |

the average logarithmic ratio between the expected change in quality prediction and the actual change in quality prediction caused by adversarial video as

$$
R = \frac{1}{S} \sum_{i=1}^{S} \log \left( \frac{\left| f\left( \left( \{\mathbf{I}_x\}_{x=1}^{X} \right)_i \right) - b\left( \left( \{\mathbf{I}_x\}_{x=1}^{X} \right)_i \right) \right|}{\left| f\left( \left( \{\mathbf{I}_x\}_{x=1}^{X} \right)_i \right) - f\left( \left( \{\mathbf{I}_x^{\text{adv}}\}_{x=1}^{X} \right)_i \right) \right|} \right),
\tag{6}
$$

where the definition of $f$ and $b$ can be referred to Section 3.1, $S$ denotes the number of videos in the test set. A higher value of $R$ indicates better model robustness. With the above three metrics, the prediction accuracy and robustness against adversarial attacks can be comprehensively evaluated.

**Implementation details:** We download the authors' source code [26; 39; 49; 31], re-train the models and produce adversarial videos on the testing set. Since the estimated quality scores' scale among different NR-VQA models is different, we need to rescale the boundary. The rescaling details can be referred to in the appendix. For a specific dataset, the median of the MOS of the selected 50 videos is regarded as the threshold to decide whether a video is of high quality or low quality in Eq. (3). As for white-box attack, Adam optimizer [47] is applied, wherein the step size $\beta$ is initialized as $3 \times 10^{-4}$. $K$ and $T$ are set to 30 and 1. The pixel-level $L_2$ norm of perturbations is constrained within 1/255. As for black-box attack, $N$, $T$, and $\gamma$ are set to 300, 1, 5/255. Both $h$ and $w$ are set to 56.

## 4.2 Performance evaluations under white-box setting

In this part, the performance of different NR-VQA models is evaluated under white-box setting, and the results under both pixel-level $L_2$ and $L_\infty$ norm constraint are given in Tables 1 and 2, respectively. It can be observed that the State-of-the-Art (SOTA) NR-VQA models are completely disabled by adversarial attack. Specifically, such attack under both pixel-level $L_2$ and $L_\infty$ norm constraint bring similar attack effects and lead to performance degradation of 194% and 195% on SRCC averaged on five datasets, demonstrating the vulnerabilities of these NR-VQA models. Meanwhile, the $R$ values on all datasets approximate to 0, indicating that the adversarial attack with our proposed Score-Reversed Boundary Loss can push the estimated quality score towards the target boundary. Note that in white-box setting, the pixel-level $L_2$ and $L_\infty$ norm of perturbations are constrained within 1/255 and 3/255, indicating that the degree of perturbation is kept on an extremely low level.

## 4.3 Performance evaluations under black-box setting

In this part, the performance of different NR-VQA models is evaluated under black-box setting, and the results are given in Table 3. It can be observed that the SRCC and PLCC of the attacked

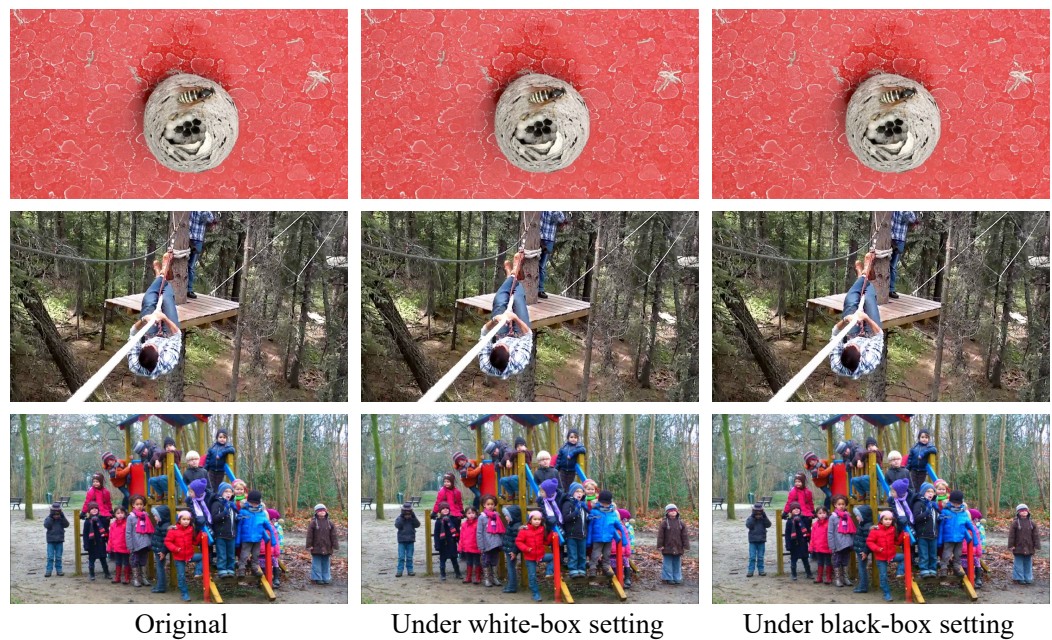

|  Original | Under white-box setting | Under black-box setting |

Figure 3: The original video frames and the corresponding adversarial video frames.

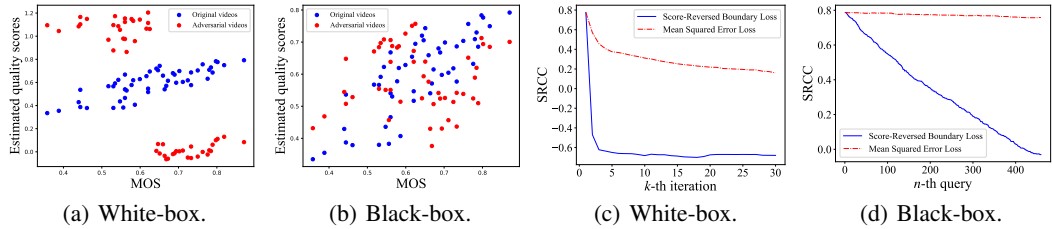

(a) White-box.    (b) Black-box.    (c) White-box.    (d) Black-box.

Figure 4: The attack effect on estimated quality scores ((a) and (b)) and the query efficiency with increasing iterations and number of queries ((c) and (d)).

VSFA, MDTVSFA, and TiVQA are around 0, indicating that the attack effect on these models is quite remarkable. Although the attack effect on BVQA-2022 is less significant, it still brings performance degradation of 0.6449 on SRCC and 0.6254 on PLCC on the KoNViD-1k dataset. Such results indicate that the current VQA systems are not robust against adversarial attacks, and can be severely disturbed in the black-box setting. Note that according to Eq. (5), the theoretical upper bound of the pixel-level $L_2$ norm and $L_\infty$ norm of the proposed patch-based random search method are both 5/255. Interestingly, in practical deployment, we find that only about 3/5 patches are injected with perturbations, and the pixel-level $L_2$ norm is around 3/255. In Fig. 3, the examples of the original video frames and the corresponding adversarial video frames under both white-box and black-box settings are presented, which are indistinguishable from their original counterparts by human eyes.

### 4.4 Ablation study

In this part, ablation studies are conducted from four aspects. Firstly, the effectiveness of the proposed Score-Reversed Boundary Loss is verified under black-box and white-box settings. The experiments are conducted via attacking VSFA on the KoNViD-1k dataset, and the results are given in Fig. 4. From Fig. 4(a) and 4(b), it can be seen that minimizing the Score-Reversed Boundary Loss achieves satisfied attack effect. Specifically, in the white-box setting, the quality scores estimated by the target NR-VQA model basically reach the target boundary. From Fig. 4(c) and 4(d), it can be observed that compared with the MSE Loss, the Score-Reversed Boundary Loss shows superiority in the query efficiency. In particular, in the white-box setting, the adversarial attack with the Score-Reversed Boundary Loss can completely disable the NR-VQA model with just a few iterations.

Table 4: Performance evaluations of SRCC for patch-based and pixel-based methods.

| $n$-th query | KoNViD-1k | | LSVQ-Test | |
|---|---|---|---|---|
| | patch | pixel | patch | pixel |
| 100 | 0.5390 | 0.7850 | 0.4392 | 0.7793 |
| 300 | -0.0305 | 0.7831 | 0.2386 | 0.7747 |
| 500 | – | 0.7828 | – | 0.7738 |
| 1000 | – | 0.7823 | – | 0.7575 |

Table 5: Performance evaluations of SRCC with different values of $T$.

| $T$ | KoNViD-1k | | LSVQ-Test | |
|---|---|---|---|---|
| | White | Black | White | Black |
| 1 | -0.7003 | -0.0305 | -0.6594 | 0.2386 |
| 2 | -0.6972 | 0.1318 | -0.6667 | 0.2579 |
| 4 | -0.6747 | 0.2121 | -0.6715 | 0.3171 |
| 8 | -0.6964 | 0.2747 | -0.6768 | 0.3482 |

Table 6: Performance evaluations of motion embeddings on BVQA-2022.

| Motion embeddings | KoNViD-1k | | LSVQ-Test | |
|---|---|---|---|---|
| | SRCC | PLCC | SRCC | PLCC |
| ✗ | -0.1779 | -0.1465 | 0.1917 | 0.2641 |
| ✓ | 0.1978 | 0.2215 | 0.3355 | 0.4240 |

Table 7: Performance evaluations of motion embeddings on VSFA.

| Motion embeddings | KoNViD-1k | | LSVQ-Test | |
|---|---|---|---|---|
| | SRCC | PLCC | SRCC | PLCC |
| ✗ | -0.0305 | 0.0586 | 0.2386 | 0.2413 |
| ✓ | 0.1858 | 0.2687 | 0.4358 | 0.4536 |

Secondly, our patch-based random search method is compared with the pixel-based random search method SimBA [24] under black-box setting. The experiments are conducted via attacking VSFA on the KoNViD-1k and LSVQ datasets, and the results are given in Table 4. We can see that compared with the pixel-based method, our method shows significant improvements in both attack effect and query efficiency. Specifically, our method merely consumes 300 queries to completely disable the NR-VQA model, while the pixel-based method barely has any attack effect with 1000 queries.

Thirdly, the number of frames optimized in one round, i.e., $T$, is investigated under both black-box and white-box settings. The experiments are conducted via attacking VSFA on the KoNViD-1k and LSVQ datasets, and the results are given in Table 5. From the perspective of the white-box setting, different values of $T$ have little impact on the attack effect. However, in the black-box setting, smaller $T$ corresponds to stronger attack effect. The reason is that different frames within one video have different characteristics, and thus optimizing the perturbations with smaller $T$ can bring the potential of their individual differences to full play. Considering the attack effect, $T$ is set to 1 in our method.

Fourthly, the impact of motion embeddings on the robustness of the model is explored under black-box setting. Note that BVQA-2022 applies motion embeddings, while VSFA does not. In Table 6, the original BVQA-2022 and the BVQA-2022 without motion embeddings are compared. In Table 7, the original VSFA and the VSFA with motion embeddings are compared. It can be observed that the motion embeddings play an important role in improving the robustness of NR-VQA models against adversarial attacks. Specifically, the motion embeddings bring SRCC performance improvement of 0.2598 for BVQA-2022 and 0.2068 for VSFA. One possible reason is that the motion embeddings correspond to the inter-frame information in temporal domain. Therefore, the adversarial impact of perturbations added within frames may be weakened, resulting in an improvement of robustness.

## 5 Conclusions

In this paper, we fully investigate the robustness of NR-VQA models against adversarial attacks, and propose the Score-Reversed Boundary Loss for the attack problem. Extensive experiments have been conducted under both white-box and black-box settings, and the following conclusions can be drawn. Firstly, the existing SOTA NR-VQA models are vulnerable to adversarial attacks, which brings great challenges to build reliable and practical assessment systems. Secondly, the attack effect and query efficiency are influenced by different components, including the Score-Reversed Boundary Loss, the patch-based random search method, and the number of frames optimized in one round. Thirdly, motion embeddings have a positive impact on increasing the robustness of NR-VQA models, which may be the key for developing robust NR-VQA models. These findings present a promising direction for NR-VQA. Future work may include the following aspects. Firstly, it is desired to develop the learning-based attack method, i.e., apply a generator to produce perturbations to fool the NR-VQA models. Secondly, more characteristics, such as the information in the temporal domain, should be explored to improve the robustness of NR-VQA models.

## Acknowledgments

This work was supported in part by the National Natural Science Foundation of China under Grants 62272116 and 62002075, and in part by the Guangdong Basic and Applied Basic Research Foundation under Grant 2023A1515011428. The authors acknowledge the Network Center of Guangzhou University for providing HPC computing resources.

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

## A  Rescaling boundary

Due to the differences of the estimated quality scores' scale among different NR-VQA models, we rescale the boundaries of 0 and 1 to the scale of the estimated quality scores for different NR-VQA models as

$$\text{Scaled\_boundary} = \frac{\text{boundary} - \text{MOS.mean}}{\text{MOS.std}} * \text{EST.std} + \text{EST.mean}, \tag{7}$$

where MOS.mean and MOS.std denote the mean and variance of the normalized MOS in the selected 50 videos, respectively. EST.mean and EST.std denote the mean and variance of the estimated quality scores in the selected 50 videos, respectively. Note that in our proposed adversarial attack, the boundary is set according to a specific original video's MOS and a pre-defined threshold determined by the median of MOS within the batch. In practice, these two conditions may not be available to the attacker, and the solutions are as follows. The first condition is to obtain a specific video's MOS. It is reasonable to manually annotate the video, as it did in the adversarial attack in classification task in the case of unaware of the ground-truth label. As an alternative approach, we can also apply a well-trained VQA model to obtain the estimated quality score, which can be regarded as the approximated MOS. The second condition is to obtain the threshold. We can apply the approaches in the first condition to obtain the approximate MOS for each video within the dataset, and then calculate the median value as the threshold. Besides, considering the workload for processing huge number of videos within the batch, the threshold can also be set to the average of the upper bound and lower bound of the scoring range for the dataset.

## B  Ablation study of perturbation on a few frames

The adversarial effect with respect to the number of frames to be perturbed is investigated under both black-box and white-box settings. The experiments are conducted via attacking VSFA on the KoNViD-1k and LSVQ-Test datasets, and the results are given in Tables 8 and 9, respectively. It can be observed that as the proportion of perturbed frames decreases, the attack effect is degraded under both white-box and black-box settings. Specifically, due to the difficulties of black-box attack, there is barely any attack effect when only 1/10 of frames are perturbed. By contrast, under the white-box setting, the NR-VQA model can still be fooled when a small number of frames are perturbed.

Table 8: Performance evaluations for different ratios of perturbed frames under white-box case.

| Ratio | KoNViD-1k | | LSVQ-Test | |
|---|---|---|---|---|
| | SRCC | PLCC | SRCC | PLCC |
| 0 | 0.7882 | 0.8106 | 0.7865 | 0.7972 |
| 1/10 | -0.2692 | -0.2332 | -0.2262 | -0.1813 |
| 1/5 | -0.5980 | -0.4896 | -0.4985 | -0.4373 |
| 1/2 | -0.6318 | -0.7585 | -0.6069 | -0.5992 |
| all | -0.7198 | -0.8135 | -0.6723 | -0.6765 |

Table 9: Performance evaluations for different ratios of perturbed frames under black-box case.

| Ratio | KoNViD-1k | | LSVQ-Test | |
|---|---|---|---|---|
| | SRCC | PLCC | SRCC | PLCC |
| 0 | 0.7882 | 0.8106 | 0.7865 | 0.7972 |
| 1/10 | 0.6836 | 0.7511 | 0.7097 | 0.7378 |
| 1/5 | 0.6199 | 0.7127 | 0.6597 | 0.7113 |
| 1/2 | 0.4287 | 0.5447 | 0.5050 | 0.6013 |
| all | -0.0305 | 0.0586 | 0.2386 | 0.2413 |

## C  Limitation and potential societal impact

In this paper, we mainly attack the VQA models in the spatial domain, which makes it difficult to achieve an ideal attack effect against models with rich temporal information, such as BVQA-2022. Meanwhile, optimization-based attack strategies need to generate specific perturbations for each video, which have high requirements on computing resources. Secondly, according to our work, people could launch deliberate attack on VQA models, which may reduce the users' confidence of experiencing the VQA systems. In spite of this, we believe that in order to build a reliable and practical assessment system, it is of great necessity to evaluate their robustness. Therefore, this paper can raise awareness of vulnerability in existing VQA models, which greatly outweighs its risk.

