# OpenReview forum: "Vulnerabilities in Video Quality Assessment Models: The Challenge of Adversarial Attacks"
_NeurIPS.cc/2023/Conference — NeurIPS 2023 spotlight_

### Official Review · Reviewer_NoLq · 2023-06-08

**Soundness:** 3 good
**Presentation:** 3 good
**Contribution:** 3 good
**Rating:** 8
**Confidence:** 5

**Summary:**

This paper studies the adversarial robustness of NR-VQA models. A new loss function called Score-Reversed Boundary Loss is designed to push the  estimated quality score of adversarial video far away from its MOS towards a specific boundary. It also proposes a patch-based random search method for black-box attack. The impercetibility of adversarial video is constrained by a L2-constraint.

**Strengths:**

+ It establishes the first work on evaluating the adversarial robustness of NR-VQA models with reasonable motivations.
+ Both white-box and black-box attack methods are designed.
+ Rich experiments are conducted on mainstreaming VQA datasets to evaluate the adversarial robustness of multiple NR-VQA models.
+ Insightful analysis of the adversarial robustness of specific design choices of NR-VQA models are provided.


**Weaknesses:**

- The definitons of f_{e} and f_{d} are a bit confusing. In my view, the parameters of NR-VQA models are not changed during the adversarial attack, so it would be better to simply define a NR-VQA model as f, then f(I) and f(I_{adv}) will the estiamted quality of the original and perturbed video, respectively. It's better to denote the disturbed score using another symbol without subscript.

- The disturbed score is actually a target score the median MOS of videos from a VQA dataset is set as the threshold to determine whether a video is of high quality or low quality. However, there won't be a dataset with ground-truth MOS before the adversarial attack in may cases. It is not clear how to set the boundary given an arbitrary video with or without MOS in practice.

- The L2-norm constraint is used to ensure the invisibility of perturbation. Why not using other L-norm contrainst is not explained.

-  Do the authors rescale the quality predictions of all NR-VQA models within the range [0, 1] using nonlinear mapping?

**Questions:**

1, Is it possible use other FR-IQA as the fidelity measure as in [42]?

2, Is it possible to only perturb a few frames (not all of them as done in the proposed method) to fool an NR-VQA models?

**Limitations:**

Not applicable.

---

> ### Author Rebuttal · Authors · 2023-08-10
>
> Thank you for the review. We address your mentioned weakness and questions below. In this response, the performance of VQA models is evaluated by SRCC, PLCC, and $R$. Due to the limited space in the response, only SRCC is listed in the following Tables. Note that all these three metrics exhibit similar results in our experiment.
>
> **Regarding the definitons of $f_e$ and $f_d$ :** Thanks for the valuable comment. To avoid misunderstanding, $f_e$ is modified as $f$, which denotes the estimated quality score of the target VQA model. And $f_d$ is modified as $b$, which denotes the disturbed quality score.
>
> **Regarding how to set the boundary in practice:** Thanks for the insightful comment. In our proposed adversarial attack, the boundary is set according to a specific original video's MOS and a pre-defined threshold determined by the median of MOS within the batch. In practice, these two conditions may not be available to the attacker, and the solutions are as follows. The first condition is to obtain a specific video's MOS. It is reasonable to manually annotate the video, as it did in the adversarial attack in classification task in the case of unaware of the ground-truth label. As an alternative approach, we can also apply a well-trained VQA model to obtain the estimated quality score, which can be regarded as the approximated MOS. The second condition is to obtain the threshold. We can apply the approaches in the first condition to obtain the approximate MOS for each video within the dataset, and then calculate the median value as the threshold.
> Besides, considering the workload for processing huge number of videos within the batch, the threshold can also be set to the average of the upper bound and lower bound of the scoring range for the dataset.
>
> **Regarding using other $L_p$ norm to ensure the invisibility of perturbation:** Thanks for the insightful comment. In our original manuscript, both $L_2$ and $L_\infty$ norm are utilized to ensure the imperceptibility of perturbation under black-box setting. Specifically, in the optimization process, each frame is divided into small patches, which are added with a universal perturbation map once and only once whose element's value is $-\gamma$ or $+\gamma$.
> Therefore, the $L_\infty$ norm of perturbations is mathematically guaranteed to be no greater than $\gamma$.
> In the case of white-box setting, we leverage the Projected Gradient Descent (PGD) to restrict the pixel-level $L_2$ norm.
> We supplement the experiments of
> applying PGD to restrict the $L_\infty$ norm, and the results are given in Table 1.
> It can be seen that the restrictions of $L_2$ or $L_\infty$ bring similar attack effect under white-box setting. In the experiments of the original manuscript, the values of $L_\infty$ norm under both white-box and black-box settings are given, which are 3/255 and 5/255, respectively. Besides, in the supplementary PDF, the example frames of the adversarial videos are given, wherein no modification traces can be seen. As suggested by the reviewer, we will more clearly illustrate the application of different $L_p$ norm in ensuring the invisibility of perturbation in the paper.
>
> Table 1. Performance evaluations on VSFA.
> ||SRCC||
> |-|-|-|
> ||KoNViD-1k|LIVE-VQC|
> |$L_2$ constraint|-0.7003|-0.7610|
> |$L_\infty$ constraint|-0.7190|-0.7474|
> |Without attack|0.7882|0.7665|
>
> **Regarding rescaling the boundaries for different NR-VQA models:** Thanks for the valuable comment. We rescale the boundaries of 0 and 1 to the scale of the estimated quality scores for different NR-VQA models as follows.
> Firstly, given a video dataset, we sample a video batch to launch adversarial attack.
> Then, we normalize each sampled video's  MOS into [0,1], and then calculate the mean ($MOS.mean$) and variance ($MOS.std$) according to all normalized MOS within the batch.
> Afterwards, given this video batch and a target NR-VQA model, we obtain the estimated quality scores, and then calculate the mean ($EST.mean$) and variance ($EST.std$) according to all estimated scores within the batch.
> Finally, the boundaries can be rescaled from the scale of the normalized MOS to the scale of the estimated quality score as
> $$scaled\ boundary=\frac{boundary-MOS.mean}{MOS.std}*EST.std+EST.mean$$
> where the boundary equals 0 or 1. For clear illustration, we will clarify the rescale strategy in the experimental setup.
>
> **Regarding using other FR-IQA as the fidelity measure:** Thanks for the valuable comment. As suggested by the reviewer, the attack effect using other FR-IQA as the fidelity measure is verified under the white-box setting. The experiments are conducted via attacking VSFA, and the results are given in Table 2. It can be observed that using other FR-IQA as the fidelity measure brings a similar attack effect to our method.
>
> Table 2. Performance evaluations on VSFA.
> ||SRCC||
> |-|-|-|
> |FR-IQA|KoNViD-1k|LIVE-VQC|
> |Chebyshev|-0.7120|-0.7409|
> |SSIM|-0.7100|-0.7527|
> |LPIPS|-0.6930|-0.7418|
> |DISTS|-0.6989|-0.7501|
> |None|-0.7003|-0.7610|
>
> **Regarding only perturbing a few frames to fool an NR-VQA model:** Thanks for the valuable comment. Adversarial effect with respect to the number of frames to be perturbed is investigated. The experiments are conducted via attacking VSFA, and the results are given in Tables 3 and 4. It can be observed that there is barely any attack effect when only 1/10 of frames are perturbed under black-box setting. By contrast, under the white-box setting, the NR-VQA model can still be fooled when a small number of frames are perturbed.
>
> Table 3. Performance evaluations under white-box setting.
> ||SRCC||
> |-|-|-|
> |Perturbation ratio|KoNViD-1k|LIVE-VQC|
> |0|0.7882|0.7665|
> |1/10|-0.2692|-0.1872|
> |1/5|-0.5980|-0.4757|
> |1/2|-0.6318|-0.6374|
> |all|-0.7003|-0.7610|
>
> Table 4. Performance evaluations under black-box setting.
> ||SRCC||
> |-|-|-|
> |Perturbation ratio|KoNViD-1k|LIVE-VQC|
> |0|0.7882|0.7665|
> |1/10|0.6836|0.6403|
> |1/5|0.6199|0.5604|
> |1/2|0.4287|0.2441|
> |all|-0.0305|-0.1605|

---

> > ### Comment · Reviewer_NoLq · 2023-08-11
> > **Post-rebuttal comments**
> >
> > The authors have addressed my concerns well in the rebuttal, I am glad to raise my rating.
> >
> > To summarize the contributions of this work, I reach three major aspects:
> >
> > First, this work makes one of the first attempt to explore adversarial robustness of the NR-VQA problem. Although it is conceptually a seemingly direct extension of previous work on NR-IQA adversarial robustness, the problem itself is non-trivial in practice due to the high computational complexity of VQA models and the complex JND mechanism of videos.
> >
> > Second, this work explores both white-box and black-box adversarial attacks methods, leading to a more comprehensive methodological framework, which enjoys favorable potential in practical scenarios. The effectiveness of the proposed adversarial attack is also verified in the rebuttal, where attacking only a few frames can successfully fool the VQA models, facilitating a promising direction to handle the computational complexity problem.
> >
> > Third, my major concern about how to set the boundary in initial review has been addressed well. The authors have provided a promising alternative manner when the ground-truth MOSs are not available.

---

### Official Review · Reviewer_xq3Y · 2023-06-26

**Soundness:** 3 good
**Presentation:** 2 fair
**Contribution:** 3 good
**Rating:** 5
**Confidence:** 4

**Summary:**

This paper proposes a patch-based random search method for white-box attacks and black-box attacks for NR-VQA models. A Score-Reversed Boundary Loss is proposed to push the adversarial video's estimated quality score far away from its MOS towards a specific boundary. And the robustness of NR-VQA models is evaluated.

**Strengths:**

1. This paper investigates the robustness of SOTA NR-VQA models against adversarial attacks.

2. The Score-Reversed Boundary Loss gives a useful guide for the decrease of  SROCC and PLCC.

**Weaknesses:**

- The comparsion with existed methods should be contrained, like [Shumitskaya, BMVC, 2022].

[Shumitskaya, BMVC, 2022] Ekaterina Shumitskaya, Anastasia Antsiferova, and Dmitriy S Vatolin. Universal perturbation attack on differentiable no-reference image- and video-quality metrics. In 33rd British Machine Vision Conference 2022, BMVC 2022, London, UK, November 21-24, 2022. BMVA Press, 2022. URL https://bmvc2022.mpi-inf.mpg.de/0790.pdf.


**Questions:**

- What is the patch selection strategy for Line 8 in Algorithm 2? What is the generated strategy for $\mathbf{m}_n$ in Line 10 in Algorithm 2? And what is the visibility of the perturbation of the attacked images?
- Whether four NR-VQA models are strictly trained on the training set, and the samples to be attacked are selected in the test set? The re-trained progress is not mentioned in the paper.

**Limitations:**

No limitation and societal impact is discussed.

---

> ### Author Rebuttal · Authors · 2023-08-10
>
> Thank you for the review. We address your mentioned weakness, questions, and limitations below.
>
> **Regarding the comparison with [1]:** Thank you very much for reminding us of [1] which has been neglected in searching for the related works. We will include this paper in the related work in the revised manuscript.
> The differences between our work and [1] mainly come from three aspects.
> Firstly, [1] applies white-box attack, while our method considers the more practical attack scenario of black-box attack.
> Secondly, the goals of adversarial attack on VQA models are different.
> [1] aims to merely increase the estimated quality score outputted by the target model. By contrast, our method aims to mislead the score towards the direction farthest away from its MOS and thus can cause more intense disturbance on VQA models, which is more in line with the requirement of adversarial attack.
> Thirdly, [1] does not consider the visual quality during optimization, and thus the adversarial videos suffer from obvious artifacts. By contrast, to obtain satisfied imperceptibility, our method introduces the JND constraint during optimization. As a result, from the supplementary PDF, the adversarial videos are indistinguishable from their original counterparts by human eyes.
>
> [1] E. Shumitskaya et al., ``Universal perturbation attack on differentiable no-reference image- and video-quality metrics,'' in *British Machine Vision Conference*, 2022.
>
> **Regarding the patch selection strategy for Line 8 in Algorithm 2:** Thanks for the valuable comment. In Algorithm 2, the height and width of a video frame are denoted as $H$ and $W$, respectively. The height and width of a patch are denoted as $h$ and $w$, respectively. Therefore, a three-channel frame can be divided into $\lfloor \frac{H}{h} \rfloor \times \lfloor \frac{W}{w} \rfloor \times 3$ non-overlapping patches. All these patches within one frame would be selected once and only once in $N$ queries during optimization. Specifically, Line 8 in Algorithm 2 indicates that in each query, $ \max \left( \lfloor \lfloor \frac{H}{h} \rfloor \times \lfloor \left. \left. \frac{W}{w} \rfloor \times 3 / N \rfloor \right. \right. , 1 \right)$ patches within one frame would be selected for perturbation.
>
>
> **Regarding $\mathbf{m}_n$ in Algorithm 2:** Thanks for the valuable comment. In Algorithm 2, $\mathbf{m}_n$ refers the universal perturbation in the $n$-th query.
> Note that one universal perturbation map is added to all selected patches in all frames that optimized in one query together.
> By this means, the query resources can be largely reduced. Specifically, the dimension of the universal perturbation is the same as the dimension of the selected patch, and the element of universal perturbation map is independently sampled from a discrete set $\\{ -\gamma,  +\gamma \\}$.
>
> **Regarding the visibility of the perturbation of the attacked videos:** Thanks for the valuable comment. In the supplementary PDF, we select some example frames of the generated adversarial videos under both white-box and black-box settings, wherein no modification traces can be seen.
>
> **Regarding the re-trained process of the NR-VQA models:**
> Thanks for the valuable comment. In the re-trained process, we download the authors' source code, re-train the model and produce adversarial videos according to the following settings.
> For a specific dataset, the videos are randomly split into a training set, validation set, and testing set.
> The VQA models are trained on the training set, and validated on the validation set.
> Afterwards, with the well-trained VQA models, the adversarial videos are generated on the testing set.
> As suggested by the reviewer, we will add the description of the re-trained progress in the experimental setup.
>
> **Regarding the limitation and potential societal impact of our work:** Thanks for the valuable comment. The limitation and societal impact of our work are as follows. Firstly, our method mainly attacks the VQA models in the spatial domain, which makes it difficult to achieve an ideal attack effect against models with rich temporal information, such as BVQA-2022. Meanwhile, optimization-based attack strategies need to generate specific perturbations for each video, which have high requirements on computing resources. Secondly, according to our work, people could launch deliberate attack on VQA models, which may reduce the users’ confidence of experiencing the VQA systems. However, we believe that in order to build a reliable and practical assessment system, it is of great necessity to evaluate their robustness. Therefore, this paper can raise awareness of vulnerability in existing VQA models, which greatly outweighs its risk. As suggested by the reviewer, we will clarify our work's limitations and potential societal impact in the text for a clear illustration.

---

> > ### Comment · Reviewer_xq3Y · 2023-08-14
> > **Thank you for the response**
> >
> > Thank you for the response.

---

### Official Review · Reviewer_mjyw · 2023-07-04

**Soundness:** 2 fair
**Presentation:** 3 good
**Contribution:** 2 fair
**Rating:** 7
**Confidence:** 4

**Summary:**

The paper studies the adversarial robustness of No Reference Video Quality Assessment Algorithms. The paper introduces a new loss function, "Score-Reversed Boundary Loss," and demonstrates its usefulness in designing adversarial attacks for NR VQA algorithms. Overall the paper is well written, contains adequate evaluations, and a decent amount of ablation studies.

**Strengths:**

1. Introduces the new loss function "Score-Reversed Boundary Loss" and demonstrate its usefulness in designing attacks for NR VQA algorithms
2. Thorough experimentation of the proposed approach on multiple NR VQA algorithms.
3. Decent amount of ablation study

**Weaknesses:**

1. Video Quality Assessment is a Perceptual Science, and JND is motivated by Vision Science principles. However, the authors base their work on the claim that strict L2 norms would be sufficient to uphold JND constraints. The work does not guarantee that the adversarial videos are under the JND constraint for a "Human Eye." In this work, human eyes have not verified the adversarial videos are indeed under the JND constraint and the attacks are effective and imperceptible. The authors should have conducted some human study to verify their JND claim of using a strict L2 norm.
2. Also, the hyperparameter T=1 set for black-box attack gives the highest scores but potentially results in some sort of temporal flickering. Something that can be verified only by human eyes and not an L2 norm constraint.

Minor Typo :
1. Line 178 pathes -> patches?


**Questions:**

Can strict L2 norms always result in upholding JND constraints?

**Limitations:**

1. The JND constraint is not based on the principles of Vision Science.

---

> ### Author Rebuttal · Authors · 2023-08-10
>
> Thank you for the review. We address your mentioned weakness, questions, and limitations below.
>
> **Regarding JND constraint based on the principles of Vision Science:** Thanks for the valuable comment. In our work, the JND constraint is designed based on the principles of Vision Science, and the design philosophy is as follows.
> Due to the properties of retinal visual cells, the human visual system (HVS) has limited resolution and can only perceive changes larger than a certain threshold [1]. The JND can be regarded as the threshold, which represents the visibility of the HVS. Therefore, it is useful to introduce the JND threshold into the orientated signal processing systems [2]. However, JND is not a fixed value but can be influenced by many factors, including brightness, contrast between the background, and individual visual sensitivity [3]. Therefore, some fields convert it into some objective metrics for simple modeling, including PSNR in video coding [4], SSIM in image/video enhancement [5], and $L_p$ norm in adversarial attacks [6]. In our attack under black-box setting, both $L_2$ and $L_\infty$ norm are utilized to restrict the perturbations below the JND threshold. In the case of white-box setting, we leverage the Projected Gradient Descent (PGD) to restrict the pixel-level $L_2$ norm. As suggested by the reviewer, we will emphasize the reasons of utilizing $L_p$ norm for modeling the JND constraint based on the principles of Vision Science in the text.
>
>
> [1] N. Jayant et al., ``Signal compression based on models of human perception,'' *Proceedings of the IEEE,* 1993.
>
> [2] J. Wu et al., ``A survey of visual just noticeable difference estimation,'' *Frontiers of Computer Science*, 2019.
>
> [3] J. Rovamo et al., ``Modelling contrast sensitivity as a function of retinal illuminance and grating area,'' *Vision Research,* 1994.
>
> [4] X. Yang et al., ``Just noticeable distortion model and its applications in video coding,'' *Signal Processing: Image Communication,* 2005.
>
> [5] X. Dong et al., ``A pixel-based outlier-free motion estimation algorithm for scalable video quality enhancement.'' *Frontiers of Computer Science,* 2015.
>
> [6] R. Pinot et al., ``Randomization matters how to defend against strong adversarial attacks,'' in *International Conference on Machine Learning,* 2020.
>
> **Regarding whether strict $L_2$ norm always upholds JND constraint:** Thanks for the insightful comment. The authors acknowledge that in extreme situations, only applying $L_2$ norm may lead to modifications exceeding the JND in a local region.
> To address the issue, $L_\infty$ norm can be introduced to model the JND constraint, which can limit each pixel's maximum modification value.
> Note that in our original manuscript, both $L_2$ and $L_\infty$ norm are utilized to optimize the perturbations under black-box setting, which can more comprehensively uphold the JND constraint. Specifically, in the optimization process, each frame is divided into small patches, which are added with a universal perturbation map once and only once whose element's value is $-\gamma$ or $+\gamma$.
> Therefore, the $L_\infty$ norm of perturbations
> is mathematically guaranteed to be no greater than $\gamma$.  In the case of white-box setting, we leverage the Projected Gradient Descent (PGD) to restrict the pixel-level $L_2$ norm.
> We supplement the experiments of
> applying PGD to restrict the $L_\infty$ norm, and the results are given in Table 1. It can be seen that the restrictions of $L_2$ or $L_\infty$ bring similar attack effect under white-box setting. In the experiments of the original manuscript, the values of $L_\infty$ norm under both white-box and black-box settings are given, which are 3/255 and 5/255, respectively. As suggested by the reviewer, we will clarify the relation between $L_2$ norm and JND constraint throughout the paper.
>
> Table 1. Performance evaluations on VSFA.
> ||SRCC||PLCC||
> |-|-|-|-|-|
> ||KoNViD-1k|LIVE-VQC|KoNViD-1k|LIVE-VQC|
> |$L_2$ constraint|-0.7003|-0.7610|-0.7938|-0.7332|
> |$L_\infty$ constraint|-0.7190|-0.7474|-0.8135|-0.7322|
> |Without attack|0.7882|0.7665|0.8106|0.7506|
>
> **Regarding the human subjective experiments:** Thanks for the valuable comment. As suggested by the reviewer, we design a set of subjective experiments to verify whether the JND constraint can be held in our method. According to [1], the details of human subjective experiments are as follows. (1) *Data collection:* We collect 24 videos from the publicly available dataset KoNViD-1k. Then, these videos are divided into two groups for white-box and black-box attacks, respectively. In this way, one group of videos contains 12 original videos and 12 adversarial videos. (2) *Test methodology:* We recruit 15 human subjects to participate in the subjective experiments. For each yes-no trial, subjects are shown an adversarial video and its corresponding original video, and then asked to select the original video.
> In total, we collect 180 opinions for both white-box and black box settings.
> Results show that the probability of correctly selecting the original video is 48.3% and 53.3% under white-box and the black-box settings, respectively. Such results are close to random guessing, demonstrating that the adversarial videos generated by our method can achieve satisfied imperceptibility. In the supplementary PDF, the example frames of the adversarial videos are given, wherein no modification traces can be seen.
>
> [1] W. Zhang et al., ``Perceptual attacks of no-reference image quality models with human-in-the-loop,'' in *Advances in Neural Information Processing Systems,* 2022.
>
> **Regarding the minor typo in line 178:** Thanks for the kind comment. We have corrected it in the text.

---

> > ### Comment · Reviewer_mjyw · 2023-08-15
> >
> > Thanks to the authors for attempting to address my concerns.
> >
> > After reviewing the comments from the other reviewers, and rebuttal comments/clarifications, I have decided to increase my original score.

---

### Official Review · Reviewer_C8CZ · 2023-07-05

**Soundness:** 3 good
**Presentation:** 2 fair
**Contribution:** 2 fair
**Rating:** 6
**Confidence:** 5

**Summary:**

The adversarial attacks against video quality assessment models are proposed in both white box and black box scenarios.

**Strengths:**

1. The problem is important and practical.
2. The attempt to avoid subjective testing is a good try.
3. The attacks seem to work as in Tables 1 and 2.

**Weaknesses:**

1. The authors claim that "the problem of constraining the distortion below the JND threshold can be converted into the problem of restricting their pixel-level L2 norm below a rather strict threshold." A simple thought experiment would reveal that this is not true. Even if the threshold is set to $1/255$, we can simply use this budget to modify only a local region of pixels to be above JND.
2. The proposed method resembles those presented in the image/video classification literature and thus lacks novelty.
3. Why not use $\ell_\infty$-norm as the primary JND constraint?
4. The JND constraint should be more mathematically or experimentally justified.

**Questions:**

1. Why $f_d(\cdot)$ needs to take the test video as input?
2. What are the motivations of Eq. (3)?
3. What does the universal perturbation mean in the black box attack?

**Limitations:**

N.A.

---

> ### Author Rebuttal · Authors · 2023-08-10
>
> Thank you for the review. We address your mentioned weakness and questions below.
>
> **Regarding the association of JND constraint in our work with $L_2$ and $L_\infty$ norm:** Thanks for the insightful comment. The authors agree that in extreme situations, only applying $L_2$ norm may lead to modifications exceeding the JND in a local region, and $L_\infty$ norm may be more appropriate to model the JND constraint.
> Note that in our original manuscript, both $L_2$ and $L_\infty$ norm are utilized to restrict the perturbations below the JND threshold
> under black-box setting. Specifically, in the optimization process, each frame is divided into small patches, which are added with a universal perturbation map once and only once whose element's value is $-\gamma$ or $+\gamma$.
> Therefore, the $L_\infty$ norm of perturbations
> are mathematically guaranteed to be no greater than $\gamma$.
> In the case of white-box setting, we leverage the Projected Gradient Descent (PGD) to restrict the pixel-level $L_2$ norm. According to the comment, we supplement the experiments of
> applying PGD to restrict the $L_\infty$ norm, and the results are given in Table 1.
> It can be seen that the restrictions of $L_2$ or $L_\infty$ bring similar attack effect under white-box setting. As suggested by the reviewer, we will emphasize the role of $L\infty$ norm in our JND constraint throughout the paper.
>
> Table 1. Performance evaluations on VSFA.
> ||SRCC||PLCC||
> |-|-|-|-|-|
> ||KoNViD-1k|LIVE-VQC|KoNViD-1k|LIVE-VQC|
> |$L_2$ constraint|-0.7003|-0.7610|-0.7938|-0.7332|
> |$L_\infty$ constraint|-0.7190|-0.7474|-0.8135|-0.7322|
> |Without attack|0.7882|0.7665|0.8106|0.7506|
>
> Besides, three experiments are conducted to justify the JND constraint in our method.
> Firstly, $L_\infty$ norm is utilized to measure the visual quality of the adversarial videos. In the experiments of the original manuscript, the values of $L_\infty$ norm under both white-box and black-box settings are given, which are 3/255 and 5/255, respectively. Secondly, we design a set of subjective experiments to verify whether the JND constraint can be held in our method. According to [1], the details of human subjective experiments are as follows. (1) *Data collection:* We collect 24 videos from the publicly available dataset KoNViD-1k. Then, these videos are divided into two groups for white-box and black-box attacks, respectively. In this way, one group of videos contains 12 original videos and 12 adversarial videos. (2) *Test methodology:* We recruit 15 human subjects. For each yes-no trial, subjects are shown an adversarial video and its corresponding original video, and then asked to select the original video.
> In total, we collect 180 opinions for both white-box and black box settings.
> Results show that the probability of correctly selecting the original video is 48.3% and 53.3% under white-box and black-box settings, respectively. Such results are close to random guessing, demonstrating the imperceptibility of the perturbations. Thirdly, in the supplementary PDF, the example frames of the adversarial videos are given, wherein no modification traces can be seen.
>
> [1] W. Zhang et al., ``Perceptual attacks of no-reference image quality models with human-in-the-loop,'' in *NeurIPS,* 2022.
>
> **Regarding the novelty of the paper:** Thanks for the valuable comment.
> Adversarial attacks have been widely studied in classification task. However, there have been few investigations of adversarial attacks in Video Quality Assessment (VQA). Their main differences are: (1) The adversarial classification aims to fool the target model to predict a wrong label, which is a discrete value. In contrast, the adversarial VQA aims to fool the target model to output the score towards the direction farthest away from its MOS, which is a continuous value.
> (2) Due to the distinctions between different tasks, existing loss functions, such as the cross-entropy applied in adversarial classification are incapable of misleading the predicted score towards the expected direction.
>
> We clarify the novelties of this work again from the perspective of imperceptibility, effectiveness, and efficiency as follows.
> (1) Adversarial attacks on NR-VQA models are first formulated as misleading the estimated quality score under the JND constraint. (2) A novel loss function called Score-Reversed Boundary Loss is designed to push the adversarial video's estimated quality score towards a specific boundary. (3) A patch-based random search method is designed for black-box attack, which can significantly improve the query efficiency in the spatial and temporal domains.  (4) This is the first work to perform the black-box adversarial attack on VQA models. Comprehensive experiments confirm the effectiveness of the proposed method.
>
> **Regarding the motivations of Eq. (3) and the input of $f_d(.)$ :** Thanks for the valuable comment. The motivations of Eq. (3) in the original manuscript are as follows.
> A well-performed adversarial attack on NR-VQA model should possess the capability of leading the estimated quality score far away from its MOS in a specific direction, i.e., misleading the NR-VQA model to assign high score to low-quality video, and vice versa. Therefore, $f_d(.)$ takes the test video as input and outputs the specific boundary based on the MOS of the test video. Afterwards, the perturbations are optimized based on the distance between the estimated quality score outputted by the target model and the specific boundary.
>
> **Regarding the universal perturbation in the black-box attack:** Thanks for the valuable comment. To reduce the query resources in the black-box attack, one universal perturbation map is added to all selected patches in all frames that are optimized in one query together. Specifically, the dimension of the universal perturbation is the same as the dimension of the selected patch, and the element of universal perturbation map is independently sampled from a discrete set $\\{-\gamma,+\gamma\\}$.

---

> > ### Comment · Reviewer_C8CZ · 2023-08-11
> > **Final comment**
> >
> > Thank the authors for handling the reviewer's comments with a detailed and convincing rebuttal. Aftering taking other reviewers' comments (and the corresponding response) into account, the reviewer is willing to uplift the rating.

---

### Author Rebuttal · Authors · 2023-08-10

According to reviewers C8CZ, mjyw, xq3Y, and NoLq, the examples of the original video frames and the corresponding adversarial video frames are presented in the supplementary PDF under both white-box and black-box settings.

---

### Decision · Program_Chairs · 2023-09-21

**Decision:**

Accept (spotlight)

**Comment:**

The paper received uniformly positive ratings from the reviewers. The authors' rebuttal further clarifies various concerns of the reviewers. All reviewers are convinced with its technical novelty and contributions. The AC reads the reviews and discussions and recommends to accept the paper.